# Maternal health and health-related behaviours and their associations with child health: Evidence from an Australian birth cohort

Kabir Ahmad[1,2¤]*, Enamul Kabir[3], Syed Afroz Keramat[1,4], Rasheda Khanam[1]

**1** School of Business, Faculty of Business, Education, Law and Arts, and Centre for Health Research, University of Southern Queensland, Toowoomba, Australia, **2** Research Unit, Purple Informatics, Dhaka, Bangladesh, **3** School of Sciences, Faculty of Health, Engineering and Sciences, University of Southern Queensland, Toowoomba, Australia, **4** Economics Discipline, Social Science School, Khulna University, Khulna, Bangladesh

¤ Current address: School of Business, Faculty of Business, Education, Law and Arts, and Centre for Health Research, University of Southern Queensland, Toowoomba, Australia

* kabir.ahmad@usq.edu.au, purple.informatics@gmail.com

**Data Availability Statement:** Data used in this study can be accessed through contacting the Longitudinal Study of Australian Children Dataverse of National Centre for Longitudinal Data, Australian

## Abstract

### Objective

This study investigates the associations between maternal health and health-related behaviours (nutrition, physical activity, alcohol consumption and smoking) both during pregnancy and up to 15 months from childbirth and children's health outcomes during infancy and adolescence (general health, presence of a chronic illness, and physical health outcome index).

### Methods

This study used Wave 1 (2004) and Wave 7 (2016) data from the Longitudinal Survey of Australian Children (LSAC). We measured mothers' general health, presence of a medical condition during pregnancy and mental health during pregnancy or in the year after childbirth. We subsequently measured the children's general health, presence of a medical condition, and physical health outcome index at ages 0–1 (infancy) and 12–13 (adolescence). Binary logistic and linear regression analyses were performed to examine the association between the mothers' health-related variables and their children's health.

### Results

Our results showed that poor general health of the mother in the year after childbirth was associated with higher odds of poor health in infants and adolescents in all three dimensions: poor general health (OR: 3.13, 95% CI: 2.16–4.52 for infants; OR: 1.39, 95% CI: 0.95–2.04 for adolescents), presence of a chronic condition (OR: 1.47, 95% CI: 1.19–1.81 for adolescents) and lower physical health score (b = −0.94, p-value <0.05 for adolescents). Our study also revealed that the presence of a chronic condition in mothers during pregnancy significantly increased the likelihood of the presence of a chronic condition in their

Government Department of Social Services. One can also email to ada@anu.edu.au requesting data access. The lead author was granted permission to the data through online application from the following web link: https://growingupinaustralia.gov.au/data-and-documentation/accessing-lsac-data.

**Funding:** This research did not receive any specific grants from any funding agencies in public, commercial or not-for-profit sectors. The corresponding author (KA) has affiliation from commercial organization, Purple Informatics (PI). KA is a consultant of the commercial affiliation, PI. The funder provided support in the form of consultancy fees for authors KA for other works but did not provide any financial support for this work. Further, the funder (PI) did not have any additional role in the study design, data collection and analysis, decision to publish, or preparation of the manuscript. The specific roles of these authors are articulated in the 'author contributions' section.

**Competing interests:** This study is a part of PhD study of the author, KA. The corresponding author's affiliation to PI does not alter the authors adherence to PLOS ONE policies on sharing data and materials. Other authors do not have any competing interests.

offspring during infancy (OR: 1.31, 95% CI: 1.12–1.54) and during adolescence (OR: 1.45, 95% CI: 1.20–1.75). The study found that stressful life events faced by mothers increase the odds of poor general health or any chronic illness during adolescence, while stress, anxiety or depression during pregnancy and psychological distress in the year after childbirth increase the odds of any chronic illness during infancy.

## Conclusions

The present study found evidence that poor maternal physical and mental health during pregnancy or up to 15 months from childbirth has adverse health consequences for their offspring as measured by general health, presence of chronic health conditions, and physical health index scores. This suggests that initiatives to improve maternal physical and mental health would not only improve child health but would also reduce the national health burden.

## 1. Introduction

Maternal chronic illness and poor general health, particularly during pregnancy and in the year after childbirth, are increasing public health concerns as they contribute to poor health outcomes for both mother and infant [1–5]. The incidence of maternal morbidity in Australia and other developed countries has been steadily rising during the past two decades [4, 6, 7]. In a Melbourne-based study, 39% of pregnant women reported having a chronic condition with long-term health implications [8]. A Brisbane based study revealed that 34% of the antenatal care receivers were overweight or obese with the implications of increased risk of hypertensive disorders, gestational diabetes and infections during pregnancy [4]. These pregnancy complications have adverse effects on child health outcomes which require further research [1–5].

Numerous studies have investigated the association between mothers' health conditions and children's health outcomes. Barker and colleagues in the UK first revealed that poor foetal health is responsible for the increasing rate of non-communicable diseases, such as cardiovascular disease and diabetes among individuals in their adult years. This phenomenon is now known as foetal origins of adult disease [9, 10]. Subsequent studies inspired by the foetal origin hypothesis have corroborated that poor maternal prenatal and postnatal physical and mental health conditions contribute to poor general health and chronic illness among offspring in later life [11–16]. Studies have also shown evidence that children's risk of experiencing chronic disease (asthma, eczema, obesity, affective disorders or behavioural problems) is associated with particular maternal health conditions, including asthma, malnutrition, obesity, gestational diabetes, and mental distress [3, 9, 13, 14, 16, 17]. Further, maternal exposure to antibiotic and anti-depressant medications during pregnancy poses significant risks of having asthma and other respiratory and allergic chronic conditions among their children [17–19].

Further studies have expanded on the foetal origins hypothesis to show that children's risk of experiencing chronic illness as adults is also associated with mothers' lifestyle and health risk behaviours during pregnancy and after childbirth. These include unhealthy dietary practices [20], lack of physical activity [20], maternal tobacco smoking [21, 22], and harmful consumption of alcohol by pregnant mothers [5, 23]. These findings indicate a growing need to better understand the prevalence of these common health conditions among mothers and how they impact their offspring's long-term health.

While many studies have linked sub-optimal foetal environments with adult disease, few have focused on children in infancy [24] or during adolescence [1, 21], and even fewer studies have addressed the health outcomes at these time points via a longitudinal survey. Adolescence

is a critical phase of life [25] and a crucial entry point for progressing toward adulthood [26]. The importance of understanding the risk factors associated with adolescent health in the context of foetal origins cannot be understated; these risk factors can offer an early warning about the health issues children and adolescents may encounter before adulthood. This study contributes to the literature by investigating the associations of maternal health (general health, chronic health conditions and mental health) and health-related behaviours (smoking, alcohol consumption, food habits and physical activity) during pregnancy or in the year after childbirth on infant and adolescent health. This study's primary aim is to identify associations between mothers' specific health conditions and health-related behaviours during pregnancy or up to 15 months from childbirth and their offspring's general health status, presence of chronic health conditions, and physical health outcome indices when they were aged 0–1 in 2004 and aged 12–13 in 2016.

## 2. Methods

### 2.1 Data

This study utilised data from the Longitudinal Study of Australian Children (LSAC). The LSAC is a nationally representative household survey that collected comprehensive data on the health, socioeconomic status, and demographic factors of birth (aged 0–1) and kindergarten (aged 4–5) cohorts and their parents or caregivers. The LSAC study followed a two-stage, stratified, clustered design using the Health Insurance Commission (HIC) Medicare database as the sampling frame. The details of the sampling design and the survey methodology have been described elsewhere [27].

  This study used data of mothers and their offspring from the birth cohort of the LSAC dataset recruited in 2004. Data on mothers' health and health-related behaviours during pregnancy or in the year after childbirth and infants' health outcomes were extracted from Wave 1. This birth cohort's health outcome data were taken again from Wave 7, when they were adolescents, thus allowing a comparison of the children's health outcomes during infancy and adolescence against their biological mothers' health conditions. Although children of the kindergarten cohort (aged 4–5) in Wave 1 became adolescents (aged 12–13) in Wave 5, infant data for this cohort is not available. Moreover, although mothers of the kindergarten cohort provided information on maternal health, those would be retrospective, as they were four years earlier and might have more recall bias. Hence, this study did not use the data of the kindergarten cohort from the LSAC survey. The birth cohort surveyed in Wave 1 in 2004 included 5107 infants, 3381 (66.2%) of whom were retained as adolescents for Wave 7 in 2016. Of these children, this study included 5019 infants and 3327 adolescents after excluding primary caregivers who were not biological mothers of the children. Details of the exclusion and attrition of families and total sample are shown in Fig 1.

### 2.2 Ethics approval and consent

The LSAC study was approved by the Australian Institute of Family Studies Ethics Committee.

  The researchers received access to the database by contacting the Longitudinal Study of Australian Children Dataverse of the National Centre for Longitudinal Data. In keeping with the national regulations, researchers may use this dataset after following certain regulations, if there is no identifiable information of individuals in the data. As there was no identifiable information of individuals in the secondary data used in this study, consent for publication is not applicable for this study.

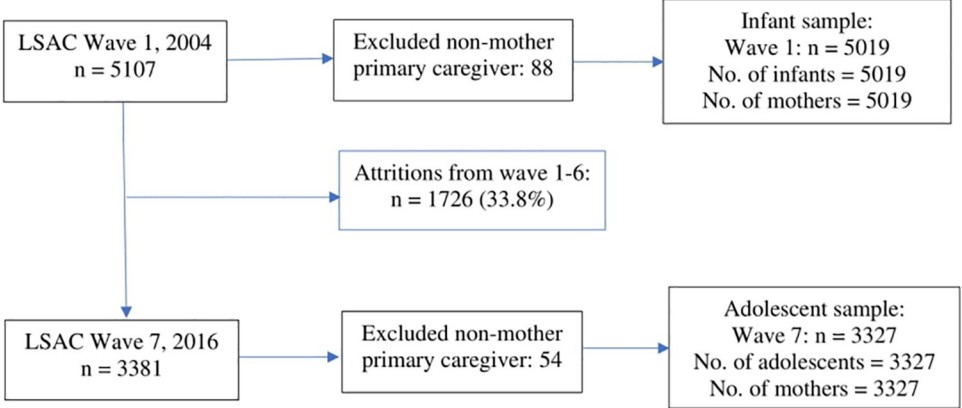

**Fig 1. Longitudinal participant numbers of the study after exclusions and attritions.**

## 2.3 Variables

**2.3.1 Dependent variables.** This study considered the following three health outcome variables:

*1. General health status.* The general health status of each child was measured via a five-point Likert scale (1 –excellent; 2 –very good; 3 –good; 4 –fair; 5 –poor) reported by the biological mother of the child. The ordinal Likert scale values were converted into dichotomous groups: children of relatively good health (excellent/very good = 0) and children of relatively poor health (good/fair/poor = 1), in keeping with previous literature [28, 29]. These two health status categories were termed 'good health' and 'poor health', respectively.

*2. Medical conditions.* A health condition or disability encountered by a child that lasted or was likely to last for six months (or more) following its initial occurrence was considered a medical condition for this study. The health problems considered were wheezing, bronchiolitis, asthma, eczema, food or digestive allergies, ear infections, hearing problems, vision problems, attention deficit disorder, other illness, and other infections. The variable was assigned a value of 1 if any medical condition was present and 0 otherwise.

*3. Physical health outcome index.* This index is a composite score calculated using the global overall health rating and scores from six-item special health care needs screening. The special health care needs screening questions were: (i) Does the child currently need or use medicine prescribed by a doctor, other than vitamins? (ii) Is this because of any medical, behavioural, or other health condition? (iii) Is this a condition that has lasted or is expected to last for at least 12 months? (iv) Does the child need or use more medical care than is usual for most children of the same age? (v) Is this because of any specific medical, behavioural, or other health condition (not just the common cold)? and (vi) Is this a condition that has lasted or is expected to last for at least 12 months? This screening was designed to assess the physical health of a child at a particular point in time, in this case, during infancy (0–1 year of age) and adolescence (12–13 years of age).

The calculation of the physical health outcome index for infants and adolescents was performed following a set of guiding principles that were developed by the members of the Outcome Index Working Group of LSAC [30] and have been used in several studies [24]. The steps are as follows: (1) calculate the average of the six-item special health care needs screening responses; (2) standardise the overall health rating variable and the average health care need screening scores so that they are weighted equally in the index; (3) multiply the standardised

overall health rating variable by −1 so that a higher value indicates a more positive outcome; (4) take the average value of the two standardised variables; (5) for adolescent samples, standardise the BMI scores and take the average of all three variables; (6) standardise the grand sum of the values of the variables and then further standardise them to a normally distributed variable with mean 100 and standard deviation (SD) 10. This final standardised variable is the physical health outcome index utilised in this study.

**2.3.2 Independent variables.** Our independent variables of interest were selected based on the existing literature on maternal health conditions and health-related behaviours as well as the variables available in the LSAC data [24, 28, 31, 32]. Details of these variables are as follows:

*Maternal health*. This study utilised the following indicators to measure maternal health: (i) general health status, (ii) presence of a medical/chronic condition, and (iii) mental health status. The health-related variables of the mother were selected from Wave 1 when their child was 0–1 year old.

In the LSAC, the mother's self-reported general health status was originally reported using a 5-point Likert scale. This measure was taken from the mothers when their children were mostly (99.3%) between 3 to 15 months old. For this study, the score was recoded into dichotomous groups: good health (0) and poor health (1), following previous literature [28, 29] and the same process was utilised in constructing the children's general health status variable.

A key explanatory variable of the prenatal period examined in this study was whether the mothers had any medical conditions during pregnancy. The conditions considered were asthma, gestational diabetes, nausea, hypertensive disorder, medical conditions for which pregnant mothers used anti-depressant, anti-allergy, or antibiotic medications, and other physical problems during pregnancy.

This study also considered three maternal mental health-related variables available in the LSAC data. These variables are: (i) had stress, anxiety, or depression during the pregnancy (Yes = 1, No = 0); (ii) the number of stressful life events in the last 12 months from the time of their interviews (no events = 0, one or more events = 1); and (iii) psychological distress experienced in the past four weeks. To measure psychological distress, mothers were asked the following questions to answer through recalling the past four weeks from interviews while their children were 3 to 15 months old. The questions were: how often they felt (a) nervous, (b) hopeless, (c) restless or fidgety, (d) that everything was an effort, (e) so bad that nothing could cheer you up, and (f) worthless. Responses were recorded on a Likert-type scale ranging from 1 ('all of the time') to 5 ('none of the time'). These scores were summed and reverse coded. Finally, the mean score of the sum of these six questions constructed the measurement scale, where a higher score represented worse mental health than a lower score. The items on this questionnaire were taken from the Kessler K6 screening scale [33].

*Mothers' health-related behaviours and risk factors*. The health-related behaviours included were the number of usual daily servings of vegetables and fruits, the number of days per week that the mother engaged in at least 30 minutes of moderate/vigorous exercise when their children were 3 to 15 months old, and food exclusion behaviour during pregnancy. Health-related risk factors included smoking frequency in a day during the first trimester of pregnancy and the number of days per week alcohol was consumed during the first trimester of pregnancy.

It is worth mentioning that this study used the independent variables of maternal physical and mental health and health-related behaviours either during pregnancy or when their children were 3 to 15 months old, as LSAC measures these variables from these timepoints. Hence, the associations derived from these variables refer the odds of child health outcomes against the respective time points' maternal health or health-related behaviours.

*Other health variables*. Further, this study utilised the variables: birth weight status ($<$2500 gm = 1, $>$2500 gm = 0) and gestational age ($<$37 weeks = 1, $> =$ 37 weeks = 0) as explanatory variables to analyse the health outcomes of the children. This is because these variables play important role in maternal and child health.

**2.3.3 Control variables.** Based on existing literature [24, 28, 29, 31], this study included several relevant socio-demographic and health-related variables of mothers, children and their families as control variables. The control variables considered in analysing the associations for both infant and adolescent regression models were as follows: gender of the child, birth type (normal/caesarean/others), immunization status of the children (completed/not completed), age of mothers ($< =$ 18 years, 19–34 years, $> =$ 35 years), home language (English, non-English), indigenous status of the child, marital status and education of the mother, income quantile of the family and remoteness of the living area (highly accessible, accessible, moderately accessible, or remote/very remote). It is expected that adolescent health outcomes could be confounded by their own health-related behaviours. Hence, for the adolescent models, the following additional control variables were considered: physical activity, smoking, alcohol consumption, and how often they ate fruits or vegetables in the previous day from the interview date. The distributions of all control variables are shown in Table 1.

## 2.4 Statistical analysis

This study applied binary logistic and linear regression models to assess the relationship between the predictor variables (mothers' health) and the three health outcome variables (children's health). The first and second binary logistic regression models assessed the children's general health status and chronic health conditions. The third model, a linear regression model, assessed the children's physical health outcome index. The study also conducted several collinearity diagnostic measures, including variance inflation factors, tolerance and eigenvalues for the independent variables specified in each regression model; no evidence of multi-collinearity was found. There was no evidence of heteroscedasticity, and the distributions were moderately normal. The socio-demographic factors were controlled for all models, and estimates were produced using population weights for all analyses. This study has sufficient statistical power regarding the sample size required to derive the statistics that represent the parameters or to avoid type 1 error of inference. Our study utilised logistic regression models for Tables 2 and 3 with 14 independent variables from the observational surveys, which require a minimum sample size of 800 in line with the recommended rules of thumb explained by Bujang et al. (2018), while our study had over 5000 observations for infants and over 3000 observations for adolescents [34]. Table 4 used the linear regression model, and its sample size was also sufficient to retain the expected level of statistical power in the analyses.

## 3. Results

Table 1 shows descriptive statistics for the mothers and children included in the samples of infants and adolescents. Among the mothers, 32.4% had poor general health, 40.2% had at least one medical condition, 15.4% had stress, anxiety, or depression during pregnancy, and 54.4% experienced one or more stressful life events in the twelve months prior to the interview at Wave 1. Among the children, 13.2% of the infants and 15.2% of the adolescents were in poor health. Further, 39.4% of the infants and 54.5% of the adolescents suffered from at least one of the selected chronic health conditions. It was found that 14.5% of the mothers smoked during pregnancy, and 20.3% had consumed alcohol in the first trimester.

Table 2 presents the associations between the mothers' general health status, chronic health conditions, mental health and health-related behaviours and their children's general health

**Table 1. Characteristics of the sampled subjects during infancy and adolescence time of the study children**\*.

| Variables | Infancy (aged 0/1, n = 5019) | | Adolescence (Follow-up of the birth cohort at age 12/13, n = 3327)* | |
|---|---|---|---|---|
| | n | % /Mean (SD) | n | % /Mean (SD) |
| **Dependent Variables** | | | | |
| General health status of children | | | | |
| Excellent/very good | 4,356 | 86.8 | 2,820 | 84.8 |
| good/fair/poor | 663 | 13.2 | 507 | 15.2 |
| Children having any chronic conditions | | | | |
| No | 3,041 | 60.6 | 1,515 | 45.5 |
| Yes | 1,978 | 39.4 | 1,812 | 54.5 |
| Physical Health Outcome Index–mean (SD)** | 5018 | 99.9 (10.0) | 3318 | 99.7 (10.3) |
| **Independent Variables** | | | | |
| General health status of the mother | | | | |
| Excellent/very good | 3,393 | 67.6 | 2,262 | 63.3 |
| Good/fair/poor | 1,626 | 32.4 | 1,065 | 32.0 |
| Mothers having any medical conditions | | | | |
| No | 3,001 | 59.8 | 1,961 | 59 |
| Yes | 2,018 | 40.2 | 1366 | 41 |
| **Mother's Mental health-related variables** | | | | |
| Mothers had stress, anxiety, or depression during pregnancy | | | | |
| No | 4246 | 84.6 | 2831 | 85.1 |
| Yes | 773 | 15.4 | 496 | 14.9 |
| Number of stressful life events mothers faced in the previous 12 months prior to the interview | | | | |
| No events faced | 2,286 | 45.6 | 1,811 | 45.6 |
| One or more events | 2733 | 54.4 | 1516 | 54.4 |
| Psychological distress–mean (SD) (K-6 depression scale) | 4,194 | 1.6 (0.6) | 2,976 | 1.6 (0.6) |
| **Mother's Health behaviour related variables** | | | | |
| Smoking frequency in 1st trimester of pregnancy | | | | |
| None | 4,290 | 85.5 | 2,873 | 86.3 |
| Less than 10 daily | 509 | 10.1 | 321 | 9.7 |
| 11+ daily | 220 | 4.4 | 133 | 4.01 |
| Number of days per week alcohol was consumed in 1st trimester | | | | |
| Zero days in a week | 3,998 | 79.7 | 2,602 | 78.2 |
| Once a week | 743 | 14.8 | 538 | 16.2 |
| More than once a week | 278 | 5.5 | 187 | 5.6 |
| Usual daily serves of vegetable intake by mother mothers (mean) | 5,019 | 2.18 (1.3) | 3,327 | 2.25 (1.3) |
| Usual daily serves of fruit intake by mother (mean) | 5,019 | 1.5 (1.2) | 3,327 | 1.5 (1.1) |
| Number of days in a week mothers engage in at least 30 minutes of moderate/vigorous exercise (mean) | 5,019 | 2.51 (1.9) | 3,327 | 2.46 (1.9) |
| Food exclusion behaviour during pregnancy | | | | |
| All food included | 2,730 | 54.8 | 1,786 | 53.7 |
| One item excluded | 1,673 | 33.6 | 1,129 | 33.9 |
| Two items excluded | 386 | 7.7 | 256 | 7.7 |
| Three or more items excluded | 228 | 4.5 | 156 | 4.6 |
| **Control Variables** | | | | |
| Sex of the children | | | | |
| Male | 2,563 | 48.9 | 1697 | 49.0 |
| Female | 2,456 | 51.1 | 1630 | 51.1 |
| Child being breastfed less than 6 months | | | | |

(*Continued*)

**Table 1.** (Continued)

| Variables | Infancy (aged 0/1, n = 5019) | | Adolescence (Follow-up of the birth cohort at age 12/13, n = 3327)* | |
|---|---|---|---|---|
| | n | % /Mean (SD) | n | % /Mean (SD) |
| No | 2,880 | 57.4 | 2,013 | 60.5 |
| Yes | 2,139 | 42.6 | 1,314 | 39.5 |
| Age of mothers | | | | |
| < = 18 | 72 | 1.4 | 39 | 1.2 |
| 19–34 | 3,625 | 72.2 | 2,402 | 72.2 |
| > = 35 | 1,322 | 26.3 | 886 | 26.6 |
| Is English spoken at home as the main language? | | | | |
| No | 634 | 12.6 | 435 | 13.1 |
| Yes | 4,385 | 87.4 | 2,892 | 86.9 |
| Is the child from an indigenous household | | | | |
| No | 4,773 | 95.1 | 3,212 | 96.6 |
| Yes | 246 | 4.9 | 115 | 3.5 |
| Marital Status when the child was born | | | | |
| With married partner | 3,540 | 70.5 | 2,452 | 73.7 |
| With defacto partner | 945 | 18.8 | 543 | 16.3 |
| Single | 534 | 10.6 | 332 | 10.0 |
| Education of Mother | | | | |
| Year 12 or less | 1,703 | 33.9 | 1,049 | 31.5 |
| Professional qualification | 1,383 | 27.6 | 906 | 27.2 |
| Graduate/ diploma | 1,617 | 32.2 | 1,149 | 34.5 |
| Post-graduate | 316 | 6.3 | 223 | 6.7 |
| Income Quantile of Weekly Family Income | | | | |
| 1st quantile | 932 | 18.6 | 569 | 17.1 |
| 2nd quantile | 948 | 18.9 | 624 | 18.8 |
| 3rd quantile | 923 | 18.4 | 634 | 19.1 |
| 4th quantile | 932 | 18.6 | 652 | 19.6 |
| 5th quantile | 920 | 18.3 | 616 | 18.5 |
| No response | 364 | 7.3 | 232 | 7.0 |
| Remoteness of area | | | | |
| Highly accessible | 2,939 | 58.6 | 1,718 | 51.6 |
| Accessible | 1,101 | 21.9 | 912 | 27.4 |
| Moderately accessible | 754 | 15.0 | 546 | 16.4 |
| Remote/very remote | 225 | 4.5 | 151 | 4.5 |
| **Adolescent's health behaviour related control variables** | | | | |
| Number of days in a week children engage in at least 30 minutes of moderate/vigorous exercise (mean) | - | - | 3,327 | 4.0 (2.2) |
| Smoked in last 12 months | | | | |
| No | - | - | 3,284 | 98.7 |
| Yes | - | - | 43 | 1.3 |
| Alcohol consumption—in the last 12 months | | | | |
| No | - | - | 3,278 | 98.5 |
| Yes | - | - | 49 | 1.5 |
| How often children had fresh fruit yesterday? | | | | |
| Not at all | - | - | 614 | 18.4 |
| Once | - | - | 972 | 29.2 |

(*Continued*)

**Table 1.** (Continued)

| Variables | Infancy (aged 0/1, n = 5019) | | Adolescence (Follow-up of the birth cohort at age 12/13, n = 3327)[*] | |
|---|---|---|---|---|
| | n | % /Mean (SD) | n | % /Mean (SD) |
| Twice or more | - | - | 1741 | 52.3 |
| How often children had fresh fruit juice yesterday? | | | | |
| Not at all | - | - | 1860 | 55.9 |
| Once | - | - | 1004 | 30.2 |
| Twice or more | - | - | 463 | 13.9 |
| How often had children cooked vegetables yesterday? | | | | |
| Not at all | - | - | 964 | 29.0 |
| Once | - | - | 1441 | 43.3 |
| Twice or more | - | - | 921 | 27.7 |
| How often children had raw vegetables yesterday? | | | | |
| Not at all | - | - | 1490 | 44.8 |
| Once | - | - | 1194 | 35.9 |
| Twice or more | - | - | 643 | 19.3 |

[*]Univariate percentages presented.

[**]Missing data excluded from analyses; Abbreviation: SD Standard Deviation.

status. We found that poor general health of mothers in the year after childbirth increased the likelihood of poor general health of their children during infancy (OR: 3.13, 95% CI: 2.16–4.52) and adolescence (OR: 1.39, 95% CI: 0.95–2.04). This study also finds that stress, anxiety, or depression during pregnancy and one or more stressful life events among mothers in the year after childbirth were associated with the poor general health of the children, but only during adolescence. The associations revealed in Table 2 were medium to large in effect size, while the magnitudes were higher in the associations of poor maternal general health and poor general health of children for infants, compared to adolescents.

Table 3 reports the associations between the mothers' health and health-related behaviours and their children's chronic health condition. The presence of a chronic condition during pregnancy significantly increased the likelihood of a chronic condition in the offspring during infancy (OR: 1.31, 95% CI: 1.12–1.54) and during adolescence (OR: 1.45, 95% CI: 1.20–1.75). The children of mothers who experienced poor general health (OR: 1.41, 95% CI: 1.16–1.72) or at least one stressful life event (OR: 1.41, 95% CI: 1.16–1.72) in the year after childbirth were more likely to suffer from a chronic illness during their adolescence than children of mothers who did not experience that illness. Mothers having stress, anxiety, or depression during pregnancy (OR: 1.21 95% CI: 0.97–1.50) and psychological distress in the past 4 weeks from the interview in the year after childbirth (OR: 1.22, 95% CI: 1.05–1.42) were also associated with the presence of a chronic health condition in the infants. However, these risk factors did not have any significant associations for the adolescents. The associations revealed in Table 3 were small to medium in effect size, while the magnitude was higher for adolescents.

Table 4 presents the associations between maternal health and health-related behaviours during pregnancy and their children's physical health outcome index scores. Stress, anxiety or depression during pregnancy was associated with lower physical health outcomes among the offspring during infancy (b = -0.93, 95% CI: -1.77 to –0.09). Children with gestational age less than 37 weeks were more likely to have lower physical health outcome score (b = -1.51, 95%

**Table 2. Maternal health and health-related behaviours and their associations with children's general health status during infancy (0–1 year of age) and adolescence (12–13 years of age).**

| Explanatory Variables | Risk of having poor general health status at: | |
|---|---|---|
| | Infancy (N = 5019) | Adolescence (N = 3327) |
| | OR (95% CI) | OR (95% CI) |
| **Mother's Health** | | |
| Poor general health status of mothers | **3.13**\*\*\***(2.16–4.52)** | **1.39**\***(0.95–2.04)** |
| Mothers having at least one of the selected medical conditions during pregnancy | 1.22 (0.86–1.74) | 1.18 (0.83–1.67) |
| Mothers having stress, anxiety or depression during pregnancy | 0.71 (0.44–1.13) | **1.57**\*\***(1.02–2.42)** |
| Any stressful life events mothers faced in 12 months prior interview | | |
| No events faced (ref.) | | |
| Yes, one or more events faced | 1.08 (0.74–1.58) | **1.94**\*\*\***(1.32–2.85)** |
| Psychological distress—(mean) (K6 depression scale) | 1.12 (0.86–1.47) | 0.74\*(0.53–1.04) |
| **Mother's Health Behaviours** | | |
| Smoking frequency in 1st trimester of pregnancy | | |
| None (ref.) | | |
| Less than 10 daily | **1.79**\*\***(1.04–3.09)** | 1.22 (0.72–2.09) |
| 11+ daily | 1.75 (0.88–3.47) | 0.62 (0.19–1.98) |
| Number of days per week alcohol was consumed in 1st trimester | | |
| Zero days in a week (ref.) | | |
| Once a week | 1.39 (0.89–2.16) | 0.76 (0.49–1.18) |
| More than once a week | **1.68**\***(0.93–3.04)** | 1.17 (0.54–2.52) |
| Usual daily serves of vegetables that mothers have | **0.87**\***(0.75–1.02)** | 0.97 (0.84–1.13) |
| Usual daily serves of fruit that mothers have | 1.06 (0.90–1.25) | **0.84**\*\***(0.71–0.99)** |
| Number of days/week mothers engage in at least 30 minutes of exercise | 1.05 (0.96–1.15) | 0.95 (0.85–1.05) |
| Food exclusion during pregnancy | | |
| All food included (ref.) | | |
| One item excluded | 0.96 (0.66–1.40) | 1.23 (0.56–1.87) |
| Two items excluded | 0.82 (0.40–1.67) | 1.02 (0.56–1.87) |
| Three or more items excluded | 1.64 (0.73–3.70) | 1.06 (0.41–2.74) |
| **Other Health Variables** | | |
| Birth weight < 2500 gm | 0.98 (0.38–2.53) | 1.56 (0.75–3.25) |
| Gestational age <37 week | 0.71 (0.32–1.55) | 0.54 (0.25–1.19) |

Notes: (i) The infancy model is are adjusted for type of birth delivery, immunisation status and breastfeeding of children and socio-demographic characteristics—age, gender of the child, education and marital status of mother, family income, language spoken at home, remoteness of the residence and for the dependent variables of other health outcome models; (ii) The adolescence model is adjusted for all the variables of infancy model and additionally adolescent health behaviour related variables mentioned in the section "Adolescent's health behaviour related control variables" of Table 1. Abbreviations: OR Odds Ratio; CI Confidence Interval; ref. Reference Category.

\*\*\* OR and 95% CI at 1% level of significance

\*\* OR and 95% CI at 5% level of significance

\* OR and 95% CI at 10% level of significance.

CI: -2.95 to -0.08) during infancy. However, these negative relationships were no longer significant during adolescence. The children of mothers who experienced poor general health in the year after childbirth were more likely to suffer from poor general health (b = -0.94, 95% CI:

**Table 3. Maternal health and health-related behaviours and their associations with children's having any of the selected medical conditions during infancy (0–1 year of age) and adolescence (12–13 years of age).**

| Explanatory Variables | Risk of having any of the selected medical conditions at: | |
| --- | --- | --- |
| | Infancy (N = 5019) | Adolescence (N = 3327) |
| | OR (95% CI) | OR (95% CI) |
| **Mother's health** | | |
| Poor general health status of mothers | 1.10 (0.92–1.31) | **1.47***(1.19–1.81)** |
| Mothers having at least one of the selected medical conditions during pregnancy | **1.31***(1.12–1.54)** | **1.45***(1.20–1.75)** |
| Mothers having stress, anxiety or depression during pregnancy | **1.21*(0.97–1.50)** | 1.11 (0.84–1.48) |
| Any stressful life events mothers faced in 12 months prior interview | | |
| No events faced (ref.) | | |
| Yes, one or more events faced | 0.91 (0.77–1.07) | 1.30**(1.07–1.57) |
| Psychological distress—(mean) (K6 depression scale) | **1.22**(1.05–1.42)** | 1.19 (0.97–1.46) |
| **Mother's health behaviours** | | |
| Smoking frequency in 1st trimester of pregnancy | | |
| None (ref.) | | |
| Less than 10 daily | 0.85 (0.64–1.11) | 1.38*(0.98–1.95) |
| 11+ daily | 0.65**(0.43–0.97) | 1.05 (0.62–1.79) |
| Number of days per week alcohol was consumed in 1st trimester | | |
| Zero days in a week (ref.) | | |
| Once a week | **1.35***(1.11–1.64)** | 1.07 (0.85–1.34) |
| More than once a week | 1.14 (0.82–1.58) | 0.91 (0.64–1.28) |
| Usual daily serves of vegetables that mothers have | 1.02 (0.95–1.09) | 1.07 (0.98–1.16) |
| Usual daily serves of fruit that mothers have | 1.02 (0.95–1.10) | **0.91**(0.83–1.00)** |
| Number of days/week mothers engage in at least 30 minutes of exercise | 0.99 (0.95–1.04) | 1.02 (0.97–1.08) |
| Food exclusion during pregnancy | | |
| All food included (ref.) | | |
| One item excluded | 0.91 (0.77–1.08) | 1.01 (0.82–1.23) |
| Two items excluded | 1.07 (0.80–1.44) | 1.31 (0,92–1.88) |
| Three or more items excluded | 1.36 (0.91–2.06) | 1.20 (0.75–1.90) |
| **Other health variables** | | |
| Birth weight < 2500 gm | 1.06 (0.70–1.61) | 1.18 (0.72–1.93) |
| Gestational age <37 week | 1.03 (0.71–1.49) | 0.74 (0.47–1.16) |

Notes: As above mentioned in Table 2. Abbreviations: OR Odds Ration; CI Confidence Interval; ref. Reference Category.

*** OR and 95% CI at 1% level of significance

** OR and 95% CI at 5% level of significance

* OR and 95% CI at 10% level of significance.

-1.89 to −0.01) during their adolescence than children of mothers who did not experience poor general health.

Among the maternal health-related behaviours, smoking habits during pregnancy had a significant negative impact on the infants' general health (OR: 1.79, 95% CI: 1.04–3.09) and the adolescents' chronic health conditions (OR: 1.38, 95% CI: 0.98–1.95). Infants whose mothers consumed alcohol more than once a week during the first trimester were 1.68 (95% CI:

**Table 4. Maternal health and health-related behaviours and their associations with children's physical health outcome index score during infancy (0–1 year of age) and adolescence (12–13 years of age).**

| Explanatory Variables | Physical health outcome index score at: | |
|---|---|---|
| | Infancy (N = 5019) | Adolescence (N = 3327) |
| | b (95% CI) | b (95% CI) |
| **Mother's health** | | |
| Poor General Health Status of Mothers | -0.40 (-1.00 to 0.21) | **-0.94**\*\*(**-1.89 to -0.01**) |
| Mothers having at least one of the selected medical conditions during pregnancy | 0.04 (-0.49 to 0.56) | 0.57 (-0.24 to 1.39) |
| Mothers having stress, anxiety or depression during pregnancy | **-0.93**\*\* **(-1.77 to -0.10)** | 0.86 (-0.37 to 2.08) |
| Any stressful life events mothers faced in 12 months prior interview | | |
| No events faced (ref.) | | |
| Yes, one or more events faced | -0.01 (-0.55 to 0.53) | 0.05 (-0.78 to 0.88) |
| Psychological distress—(mean) (K-6 depression scale) | 0.01 (-0.51 to 0.53) | -0.44 (-1.31 to 0.42) |
| **Mother's health behaviours** | | |
| Smoking frequency in 1st trimester of pregnancy | | |
| None (ref.) | | |
| Less than 10 daily | 0.50 (-0.39 to 1.39) | 0.70 (-0.78 to 2.18) |
| 11+ daily | 0.68 (-0.53 to 1.89) | -2.24 (-5.08 to 0.60) |
| Number of days per week alcohol was consumed in 1st trimester | | |
| Zero days in a week (ref.) | | |
| Once in a week | -0.02 (-0.66 to 0.62) | -0.66 (-1.65 to 0.33) |
| More than once in a week | -0.21 (-1.21 to 0.78) | 1.10 (-0.43 to 2.62) |
| Usual daily serves of vegetables that mothers have | -0.04 (-0.26 to 0.17) | -0.26 (-0.60 to 0.08) |
| Usual daily serves of fruit that mothers have | 0.07 (-0.17 to 0.30) | -0.19 (-0.58 to 0.21) |
| Number of days/week mothers engage in at least 30 minutes of exercise | **0.15**\*\* **(0.01 to 0.30)** | 0.01 (-0.24 to 0.25) |
| Food exclusion during pregnancy | | |
| All food included (ref.) | | |
| One item excluded | **-0.52**\* **(-1.09 to 0.04)** | 0.96 (0.11 to 1.80) |
| Two items excluded | -0.61 (-1.57 to 0.35) | 1.13 (-0.30 to 2.56) |
| Three or more items excluded | 0.01 (-1.36 to 1.39) | -0.72 (-3.34 to 1.90) |
| **Other health variables** | | |
| Birth weight < 2500 gm | -0.88 (-2.42 to 0.66) | 1.00 (-0.99 to 2.99) |
| Gestational age <37 week | **-1.51**\*\* **(-2.95 to -0.08)** | -0.85 (-2.87 to 1.18) |

Notes: As above mentioned in Table 2. Abbreviations: b beta coefficient; CI Confidence Interval; ref. Reference Category.

\*\*\* OR and 95% CI at 1% level of significance

\*\* OR and 95% CI at 5% level of significance

\* OR and 95% CI at 10% level of significance.

0.93–3.04) times more likely to suffer from poor general health. However, if mothers consumed alcohol even once a week, then infants of those mothers were 1.35 (95% CI: 0.93–3.04) times more likely to suffer from at least one health condition. Furthermore, if mothers avoided eggs, milk or fish during pregnancy or breastfed for less than six months, their infants were more likely to experience lower physical health outcomes during infancy. On the other hand,

more days spent with moderate to rigorous exercise by mothers increased the physical health outcome scores of the infants (Table 4).

## 4. Discussion

This study comprehensively evaluated the foetal origins of a wide range of child health outcomes from a contemporary nationally representative Australian children's birth cohort. This study found that poor maternal health status during pregnancy or in the year after childbirth was significantly associated with an increased risk of poor general health, chronic health conditions, and poor physical health outcomes in their children. These findings corroborate previous literature [1, 21, 24], showing that poor maternal health and health-related behaviours increase the odds of poor health in infants or adolescents.

Our results indicate that maternal chronic conditions during pregnancy are significantly associated with a higher likelihood of poor general health, chronic conditions or lower physical health in their offspring during infancy or adolescence. Previous studies provide evidence that chronic conditions such as obesity during pregnancy are associated with poor general health, obesity [1, 3] or heart disease [35] in their children. While the study by Callaway et al. (2006) confirms the association of maternal overweight or obesity and chronic conditions (birth defects, hypoglycaemia or jaundice) of children at their infancy, our study asserts the likelihood of having a chronic condition among the children up to their adolescence.

This study reveals that mother's poor general health in the year after childbirth was significantly associated with infants' and adolescents' poor general health. A study in the US setting corroborates our findings that mothers' rating of their general health plays a role in the maternal perception of their infant's general health status [32]. Another study conducted by Waters et al. (2000) on children aged 5 to 18 confirms our findings that mother's poor general health is associated with children's poor general health up to their adolescence [36]. However, in general, our study reveals that the odds of having poor general health during infancy is greater while the odds of this health outcome is small for adolescents. Moreover, the current study reveals the associations between mothers' poor general health and children's chronic illness and lower physical health index score, but these are significant only during the adolescent period.

The present study did not find any significant associations of low birth weight or gestational age less than 37 weeks with poor general health status or any chronic health conditions for infants and adolescents, except for a lower physical health outcome index score among infants. This may relate to the explanation by Dauglas and Gear (1976) in another study that a national survey can contribute little to the understanding of low birthweight children due to the small number of low birthweight children who suffer from problems of morbidities [37]. Our population-based study might be interpreted to indicate that low birthweight children are relatively low in numbers in relation to all births and contribute little to the total burden of children with morbidities in Australia. However, hospital-based studies with special interest groups or a subgroup of the population may have been reported with higher odds of child health outcomes for low birth weights and gestational age at birth. These two risk factors may have mediating effects on child health outcomes that need further research.

This study results reveal that stressful life events experienced by mothers in the 12 months prior to interview in the year after childbirth or the stress, anxiety or depression during pregnancy increase the odds that of poor general health among children in the long run, particularly for adolescents. On the contrary, anxiety or depression during pregnancy or psychological distress in the year after childbirth increases the odds of poor health among infants (for chronic illness or poor physical health index score), but it diminishes in the long-

term. This might relate to chronic illness during infancy which may imprint a long-term consequence of poor general health perception among parents and children, although by adolescence those children were able to self- manage the chronic illness. Our study results are in line with earlier evidence: poor perinatal maternal mental health is linked to poor infant physical health [38]; maternal psychological distress during pregnancy is associated with paediatric diseases in their offspring such as eye, ear, respiratory, digestive, and skin diseases [16] and childhood overweight [3]. Additionally, depression is associated with an unhealthy lifestyle, for example smoking and poor diet, which may also adversely affect children born to depressed mothers [39]. While these studies evaluated the association of any specific maternal mental health condition with a child health outcome, our study considers three different measures of maternal mental health either during pregnancy or when children were 3 to 15 months old, as LSAC measures these variables from these timepoints. As a result, this study has been able to evaluate the associations with confounding influence of all three types of maternal mental health. However, all these three different measurements may create a difficulty in focusing on the associations of health outcomes by specific timepoints.

The present study results showed that smoking during pregnancy was associated with children's poor health status during infancy. Similarly, a study of three birth cohorts in Finland and Denmark found that the children in each of the cohorts whose mothers smoked during pregnancy had higher hyperactivity–inattention scores compared with the children of non-smokers [21]. The findings of the present study have implications for child health outcomes in contemporary demographics of Australia, as, in 2015, almost 1 in 4 (23%) of the mothers who gave birth reported smoking during the first 20 weeks of pregnancy [7]. Our results also revealed that alcohol consumption more than once a week during the first trimester was associated with children with chronic conditions during infancy; the Western Australia study also found that Prenatal Alcohol Exposure (PAE) increases the odds of child behavioural problems [40]. These results indicate that substance use during pregnancy potentially causes impairment of the foetus.

Given that this study discussed the poor child health outcomes for women with prenatal and postnatal health conditions from a birth cohort of 2004, it might be worsened with the current Australian population. Women in Australia are continuing to give birth at an increasing age: the average age rose from 29.7 in 2005 to 30.3 in 2015 [7], with a proportion giving birth after 40 years age [41]. It was observed from a longitudinal study on women's health that the chronic conditions–hypertension, heart disease and diabetes–along with the risk factors (increasing weight and lower physical activity) were more prevalent among mid-aged (45–50 years) women [42]. Thus, adverse child physical health outcomes are more likely to increase at the population level in Australia due to the increasing maternal age of the pregnancy and increased risk of pregnancy with chronic conditions [7].

This study's main strength lies in the use of large, nationally representative and population-based contemporary birth cohort generalisable to all Australian children born in 2004. Hence the study could re-examine the foetal origins for the almost current youth population of Australia. Using the longitudinal data, this study could compare the child health outcomes for infants and adolescents against a range of maternal health indicators, including physical and mental health and health-related behaviours of mothers during pregnancy or in the year after childbirth. Another strength of the study is the adjustment of important confounders of maternal and child health. Further, for adolescent health models, this study adjusted for adolescent health behaviours related variables, which are available in LSAC, to make the estimates more reliable.

This study has several limitations that need to be discussed. As this study is not designed to assess the causal effects, we cannot ascertain the causality and its directions. Further study with

advanced techniques is required to determine the underlying mechanism to explain the associations revealed in this study. In addition, more research is necessary before any generalisations of the present study's findings to other countries can be made. Another limitation is that the general health status data for the mothers and children were self-reported and might have recall bias. However, there was no subjective bias in reporting chronic conditions or calculating the physical health outcome index. Multiple testing is another concern for this study. It tested the study sample three times to test the three hypotheses regarding the child health outcomes for each infant and adolescent sample. To address this concern, we reported the significance level of p-values and discussed the effect size where it was possible to understand the statistical power of the analysis and also discussed the scientific plausibility and supporting data from other studies to validate our study results. Further, there were different sample sizes from infants to adolescents due to dropouts. Hence, the statistical power of the analyses differs, though we got the benefit of longitudinal surveys and observing the same children. Therefore, this study results require caution to interpret the findings.

## 5. Conclusion

The present study re-examines the foetal origins hypothesis using a contemporary birth cohort dataset from Australia. The study found evidence that poor maternal physical health status and health-related behaviours during pregnancy or up to 15 months from childbirth has adverse health consequences for their children during infancy and adolescence in all three dimensions examined: poor general health, presence of chronic health conditions, and lower physical health scores. Maternal psychological distress during pregnancy also increased the odds of chronic health conditions and lower physical health scores in their offspring. These study findings emphasise the importance of improving maternal physical and mental health and promoting a healthy lifestyle during pregnancy or in the year after childbirth to improve child health. These results have policy implications for undertaking preventive measures to improve maternal health and create awareness of the importance of a healthy lifestyle during pregnancy to reduce poor health outcomes in their offspring.

## Acknowledgments

This paper uses the data from 1st and 7th Wave of Longitudinal Study of Australian Children (LSAC), initiated in 2004. The authors would like to thank the Australian Institute of Family Studies for providing the LSAC data. The authors also would like to thank Dr Barbara Harmes and Dr Gail M Ormsby for proofreading the manuscript before submission.

## Author Contributions

**Conceptualization:** Kabir Ahmad, Rasheda Khanam.

**Data curation:** Kabir Ahmad.

**Formal analysis:** Kabir Ahmad.

**Methodology:** Kabir Ahmad, Enamul Kabir, Rasheda Khanam.

**Software:** Kabir Ahmad.

**Supervision:** Enamul Kabir, Rasheda Khanam.

**Writing – original draft:** Kabir Ahmad, Syed Afroz Keramat.

**Writing – review & editing:** Kabir Ahmad, Enamul Kabir, Rasheda Khanam.

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
