## [Editor Report · Decision Letter 0]

14 Apr 2021

PONE-D-21-11263

Maternal health and health-related behaviours and their associations with child health: evidence from an Australian birth cohort

PLOS ONE

Dear Dr. Ahmad,

Thank you for submitting your manuscript to PLOS ONE. After careful consideration, we feel that it has merit but does not fully meet PLOS ONE’s publication criteria as it currently stands. Therefore, we invite you to submit a revised version of the manuscript that addresses the points raised during the review process.

We look forward to receiving your revised manuscript.

Kind regards,

Maria Christine Magnus, MPH

Academic Editor

PLOS ONE

Journal Requirements:

'  This research did not receive any specific grant from funding agencies in public, commercial or not-for-profit sectors. The corresponding author (KA) has affiliation from commercial organization, Purple Informatics (PI). KA is a consultant of the commercial affiliation, PI. However, PI did not provide any fund for this paperwork, and did not have any role in the study design, data collection and analysis, decision to publish or preparation of the manuscript for this study. Rather, this study is a part of PhD study of the author, KA. '

We note that one or more of the authors are employed by a commercial company: Purple Informatics

4. Please include a copy of Tables 1, 2, 3 and 4 which you refer to in your text on pages 9, 10 and 11.

Additional Editor Comments:

The manuscript appears to be missing all tables and figure. Please include these in the submission to allow us to evaluate your manuscript.
---

## [Author Response · Author response to Decision Letter 0]

26 Apr 2021

Additional Editor Comments:

The comments were as follows:

The manuscript appears to be missing all tables and figure. Please include these in the submission to allow us to evaluate your manuscript.

Here is our response:

Thanks so much for locating the error. Mistakenly we did not attached the tables earlier. However, in this revised submission we have presented the tables in the manuscript file. 

Please note that this study has no figure, hence no figure file was uploaded. 

Reviewers' comments:

N/A

---

## [Decision Letter · Decision Letter 1]

10 Jun 2021

PONE-D-21-11263R1

Maternal health and health-related behaviours and their associations with child health: evidence from an Australian birth cohort

PLOS ONE

Dear Dr. Ahmad,

Thank you for submitting your manuscript to PLOS ONE. After careful consideration, we feel that it has merit but does not fully meet PLOS ONE’s publication criteria as it currently stands. Therefore, we invite you to submit a revised version of the manuscript that addresses the points raised during the review process.

We look forward to receiving your revised manuscript.

Kind regards,

Maria Christine Magnus, MPH

Academic Editor

PLOS ONE

Additional Editor Comments (if provided):

I apologize for the delay in reaching a decision reagarding your manuscript. This is due to difficulties in getting external reviewers. We have now received comments from two reviewers. They have some major concerns about your manuscript. I therefore ask that you to carefully address these comments before we can reach a final decisiion about your manuscript.

Reviewers' comments:

Reviewer's Responses to Questions

**Comments to the Author**

1. If the authors have adequately addressed your comments raised in a previous round of review and you feel that this manuscript is now acceptable for publication, you may indicate that here to bypass the “Comments to the Author” section, enter your conflict of interest statement in the “Confidential to Editor” section, and submit your "Accept" recommendation.

Reviewer #1: (No Response)

Reviewer #2: (No Response)

2. Is the manuscript technically sound, and do the data support the conclusions?

Reviewer #1: Yes

Reviewer #2: Partly

3. Has the statistical analysis been performed appropriately and rigorously? 

Reviewer #1: Yes

Reviewer #2: No

4. Have the authors made all data underlying the findings in their manuscript fully available?

Reviewer #1: Yes

Reviewer #2: (No Response)

5. Is the manuscript presented in an intelligible fashion and written in standard English?

Reviewer #1: Yes

Reviewer #2: No

6. Review Comments to the Author

Reviewer #1: Manuscript Number: PONE-D-21-11263R1

Title: Maternal health and health-related behaviours and their associations with child health: evidence from an Australian birth cohort

Ahmad et al. conducted a research on a topic of interest entitled "Maternal health and health-related behaviours and their associations with child health: evidence from an Australian birth cohort." The study investigated the association between maternal general health status, mental health and chronic illness conditions, and health-related behaviours during pregnancy and in the first year postdelivery, and three health outcomes during infancy and adolescent periods. I commend the authors for their interesting work on the topic, however, I have the following concerns.

1. LSAC survey recruited infants (aged 0-1) and kindergarten kids (aged 4-5) along with their families in the first Wave (Wave 1) survey conducted in 2004. As the authors indicated those infants included in the first survey became adolescents (aged 12-13) during survey in Wave 7, similarly, kindergarten kids in Wave 1 became adolescents (aged 12-13) in Wave 5 survey conducted in 2012. However, while the present study accessed data LSAC surveys, the authors did not include a cohort of adolescents involved in 2012. This could have been due to that the kindergarten kids cohort database might not have provided information on maternal health status during pregnancy and postpartum period to be applicable for their study. This warrants some discussion in the methods section.

2. Considering the research questions studied in the study, it would be important for the authors to add some discussion related to the power and sample size of the study.

3. It was difficult to comprehend whether all or some of the key exposure variables collected maternal health information at during pregnancy or during the postpartum period in the paper. For example, as the presence of psychological distress for mothers assessed in “the past four weeks” (page 9, line 193), it was unlikely that the measures evaluated maternal mental health conditions to go back to the pregnancy period. Furthermore, the “number of stressful life events” for mothers would have covered time points back to the antepartum period. However, it was confusing when authors presented their results in one hand saying, “Psychological distress in the year after childbirth … (line 275-276) and on the other hand saying “… psychological distress during pregnancy…” (line 308). Please clarify the variables measurement and descriptions in the paper.

4. Despite the authors adjusted for key maternal demographic and birth related confounding variables, they did not provide the distributions of these maternal and birth characteristics so that someone is be able to look the distributions of these confounders in the study population.

5. While the current study provided brief information about the foetal origins of adult diseases that is used to explain some the potential associations between poor foetal health conditions and occurrence of chronic conditions later in life in the introduction section, their discussion was not sufficient to describe the potential mechanism/pathways to explain the associations revealed in their study.

6. It is also worthwhile to set a stage to explain why some of the exposures were more important for outcomes during infancy but not during adolescent period and visa versa.

7. It is also likely that adolescent health outcomes can be affected (confounded) by their health related behaviours and environmental exposures during the adolescent period. However, these potential confounders and how these factors would influence the results presented in the study were not discussed.

Reviewer #2: (No Response)

7. PLOS authors have the option to publish the peer review history of their article (what does this mean?). If published, this will include your full peer review and any attached files.

Reviewer #1: **Yes: **Gizachew Tessema

Reviewer #2: No

---

## [Author Response · Author response to Decision Letter 1]

23 Jul 2021

Thanks to the Academic Editor and the reviewers for providing constructive feedback on our paper. We have addressed all the feedback and hope the quality of this paper has now been improved substantially.

---

## [Decision Letter · Decision Letter 2]

26 Aug 2021

Maternal health and health-related behaviours and their associations with child health: evidence from an Australian birth cohort

PONE-D-21-11263R2

Dear Dr. Ahmad,

We’re pleased to inform you that your manuscript has been judged scientifically suitable for publication and will be formally accepted for publication once it meets all outstanding technical requirements.

Kind regards,

Maria Christine Magnus, MPH

Academic Editor

PLOS ONE

Reviewers' comments:

Reviewer's Responses to Questions

**Comments to the Author**

1. If the authors have adequately addressed your comments raised in a previous round of review and you feel that this manuscript is now acceptable for publication, you may indicate that here to bypass the “Comments to the Author” section, enter your conflict of interest statement in the “Confidential to Editor” section, and submit your "Accept" recommendation.

Reviewer #1: All comments have been addressed

2. Is the manuscript technically sound, and do the data support the conclusions?

Reviewer #1: Yes

3. Has the statistical analysis been performed appropriately and rigorously? 

Reviewer #1: (No Response)

4. Have the authors made all data underlying the findings in their manuscript fully available?

Reviewer #1: Yes

5. Is the manuscript presented in an intelligible fashion and written in standard English?

Reviewer #1: Yes

6. Review Comments to the Author

Reviewer #1: Thanks for the authors for responding for my comments. They have addressed my comments adequately. I have no further comments on the paper.

7. PLOS authors have the option to publish the peer review history of their article (what does this mean?). If published, this will include your full peer review and any attached files.

Reviewer #1: **Yes: **Gizachew Tessema

---

## [Editor Report · Acceptance letter]

3 Sep 2021

PONE-D-21-11263R2 

Maternal health and health-related behaviours and their associations with child health: evidence from an Australian birth cohort 

Dear Dr. Ahmad:

I'm pleased to inform you that your manuscript has been deemed suitable for publication in PLOS ONE. Congratulations! Your manuscript is now with our production department. 

Kind regards, 

on behalf of

Dr. Maria Christine Magnus 

Academic Editor

PLOS ONE